# Cell-Laden Composite Hydrogel Bioinks with Human Bone Allograft Particles to Enhance Stem Cell Osteogenesis

**DOI:** 10.3390/polym14183788

**Published:** 2022-09-10

**Authors:** Hadis Gharacheh, Murat Guvendiren

**Affiliations:** 1Otto H. York Department of Chemical and Materials Engineering, New Jersey Institute of Technology, Newark, NJ 07102, USA; 2Department of Biomedical Engineering, New Jersey Institute of Technology, Newark, NJ 07102, USA

**Keywords:** bioprinting, additive manufacturing, bone tissue engineering, bone scaffold, bone regeneration, alginate, photocurable hydrogel, rheology

## Abstract

There is a growing demand for bone graft substitutes that mimic the extracellular matrix properties of the native bone tissue to enhance stem cell osteogenesis. Composite hydrogels containing human bone allograft particles are particularly interesting due to inherent bioactivity of the allograft tissue. Here, we report a novel photocurable composite hydrogel bioink for bone tissue engineering. Our composite bioink is formulated by incorporating human allograft bone particles in a methacrylated alginate formulation to enhance adult human mesenchymal stem cell (hMSC) osteogenesis. Detailed rheology and printability studies confirm suitability of our composite bioinks for extrusion-based 3D bioprinting technology. In vitro studies reveal high cell viability (~90%) for hMSCs up to 28 days of culture within 3D bioprinted composite scaffolds. When cultured within bioprinted composite scaffolds, hMSCs show significantly enhanced osteogenic differentiation as compared to neat scaffolds based on alkaline phosphatase activity, calcium deposition, and osteocalcin expression.

## 1. Introduction

Large bone defects caused by traumatic injury, disease, infection, tumor removal, fracture, and complicated congenital malformation are difficult to treat as the size of the defect is beyond the intrinsic capacity of self-regeneration of the bone [1,2,3]. Autograft bone, i.e., bone tissue from a patient’s own body, is the gold standard for bone grafting to treat large bone defects [4,5,6,7,8]. Limitations of autograft bone, including availability of large enough bone tissue and complications in the harvesting site, such as infection, pain, and bleeding, have led to a search for alternative grafting options [5]. Bone allografts, i.e., human bone tissue from donors, and synthetic bone graft substitutes, including porous scaffolds composed of biodegradable polymers, bioceramics, and their composites, are commonly used alternatives [9,10,11,12]. Allograft bone has recently gained significant interest due to its inherent bioactivity, such as osteoconductive and osteoinductive characteristics [4,13]. Possibility of implant rejection (immunogenicity) and disease transmission are currently limiting the direct use of commercially available allograft bone tissues [14,15]. Decellularization of the allograft bone is shown to effectively reduce the risk of immunogenicity [13,16,17] and coating the allograft with different minerals is reported to enhance bone mineral deposition and functional integration of the allograft by decreasing the fibrotic tissue formation [18,19,20,21]. Despite these advancements, obtaining a large-size allograft bone that fits perfectly into the defect site, considering the size and shape of the defect, and sterilization of large-scale bone tissue without damaging structure and function remains challenging [22,23,24].

To overcome the abovementioned issues raised by the direct use of the allograft bone, allograft bone can be used as a building material to construct a scaffold that can be used as a bone graft substitute. Although conventional scaffold manufacturing techniques (casting and molding, freeze drying, salt leaching, electrospinning, etc.) provide some control on scaffold architecture, porosity, and compositional heterogeneity, additive manufacturing (AM), or 3D printing, has emerged as a revolutionary method to fabricate complex scaffolds for tissue engineering applications [23,24,25,26,27]. One of the main advantages of AM is that a patient’s medical image can be used to design and fabricate a scaffold with the correct shape of the tissue or defect [28,29,30]. Extrusion-based 3D bioprinting is an advanced AM technology, which allows direct printing of live cells alone (in the form of dense aggregates or spheroids) or supported by a hydrogel system [31,32,33]. The bioprintable live cell-containing formulation is referred to as the “bioink” [34,35,36,37]. Extrusion-based bioprinting is the most commonly used 3D bioprinting approach due to the ease of use and the availability of the bioprinters and bioinks as well as the ability to dispense large volumes of bioinks with high concentrations of live cells [38,39,40]. Moreover, the ability to simultaneously bioprint multiple bioink formulations allows fabrication of complex tissues with structural, compositional (biochemical and cellular), and mechanical heterogeneity [30,41,42,43,44,45,46,47,48,49]. These properties are also required characteristics for an ideal bone scaffold [50,51].

3D bioprinting has been the focus of bone tissue engineering, and a wide range of novel bioink formulations and bioprinting approaches has been reported to regenerate bone tissue [25,52,53,54]. Bioinks composed of allograft bone are one of the unique formulations with a significant potential to boost the bioactivity and, hence, tissue maturation and functional integration of the bioprinted tissue. There are two main approaches to develop allograft-derived bioinks. The first approach comprises development of bone mimetic cell-laden hydrogels from decellularized allograft bone tissue [55,56,57,58]. In this approach, decellularized and demineralized bone tissue is digested from a solution that can be physically crosslinked to form a stable cell-laden hydrogel at temperatures close to 37 °C [55,56,57,58]. Digested tissue can also be functionalized [59,60] or alternatively blended with other synthetic or natural polymers to form bone mimetic hydrogels [61,62]. Commonly used natural hydrogels include collagen, gelatin, silk fibroin, alginate, chitosan, and hyaluronic acid [63,64,65,66]. Synthetic hydrogels are biologically inert and do not promote cellular behavior, yet provide structural integrity and a higher degree of tunability. In the second approach, bone allograft particles are used as a bioactive filler to form cell-laden composite hydrogels. In this approach, decellularized bone tissue is processed into micron- or nano-size particles and used as an additive [67,68,69]. This approach does not require digestion, and hence demineralization of the bone tissue, and allows the allograft particles to retain their bioactivity [70]. Ratheesh et al. showed the feasibility of patient-specific bone inks by incorporating bone particles (≤500 µm) into a methacrylated gelatin bioink formulation at high particle concentrations (5%–15% *w*/*v*) [69]. They reported that both the shear thinning behavior of the inks and the mechanical strength of the bioprinted constructs increased with increasing particle concentration. Cells contained in the formulation expressed early osteogenic markers and were able to migrate and colonize the bioprinted scaffolds [69]. Kara et al. developed a bone particle reinforced composite gelatin bioink formulation by using decellularized bone particles (~100 µm) obtained from rabbit femur [68]. The stiffness and degradation rate of the scaffolds were enhanced with increasing particle content, and cells were reported to attach and proliferate around the particles as well as within the composite hydrogel [68]. In addition to 3D bioprinting, bone particles have also been incorporated in biodegradable polymers (e.g., polycaprolactone, PCL) for AM of 3D printed bone scaffolds [71,72,73,74], and in injectable colloidal hydrogels for direct injection into the defected site [75,76,77,78]. Here, we would like to note that although bioceramic particle fillers are commonly used to develop bone mimetic composite bioinks, there are limited studies focusing on bone allograft particles.

In this study, we aim to develop a novel composite bone-mimetic bioink composed of methacrylated alginate (MeALG) hydrogel filled with human bone allograft particles. Alginate is selected as the basic component of the bioink formulation due to its availability and cost, biocompatibility, and processability [79,80,81,82,83,84]. Alginate can be easily functionalized to synthesize photocurable MeALG polymer [85,86]. Methacrylate groups allow for light-induced radical polymerization (or crosslinking) to form hydrogels, and for tethering of bioactive cues in the form of cysteine-containing peptides via addition reaction [85]. Although bioceramics, such as hydroxyapatite [87], bioactive glass [88], silica [89], and calcium phosphate derivatives [90,91,92,93], are commonly incorporated into alginate to form composite hydrogels for bone tissue engineering, studies focusing on human bone allograft particles are lacking. This study aims to address this gap. Here, we report processing of human allograft tissue to form micron size particles and formulation of composite bioinks using these particles. We present detailed characterization of the rheological properties of the composite bioinks and their printability as well as mechanical behavior of the 3D bioprinted constructs. We demonstrate the use of our composite bioink formulations for bone tissue engineering by investigating human mesenchymal stem cell (hMSC) osteogenesis within 3D bioprinted constructs up to 28 days of culture, based on alkaline phosphatase and alizarin red assays as well as osteocalcin immunostaining.

## 2. Materials and Methods

### 2.1. Methacrylated Alginate (MeALG) Synthesis

Methacrylated alginate (MeALG) was synthesized as described previously [85,94]. Briefly, 0.5% (*w*/*v*) was prepared by dissolving 5 g of medium viscosity alginate (alginic acid sodium salt from brown algae, Sigma-Aldrich Inc., St. Louis, MO, USA) in 1 L of DI water. The solution was kept under magnetic stirring at 1–4 °C. Once the alginate was fully dissolved, 10 mL of methacrylate anhydride (MA, Sigma-Aldrich Inc., St. Louis, MO, USA) was added dropwise into the solution within a span of 1.5–2 h. 2 M NaOH solution (Sigma-Aldrich Inc., St. Louis, MO, USA) was simultaneously added dropwise to adjust the pH of the solution to 8–9. After the addition of the MA, pH of the mixture was maintained by gradually dripping 2 M NaOH solution for 8 h using an automated pH controller. The solution was kept at 4 °C overnight. The reaction was resumed the following day by adding 5 mL of MA while maintaining the pH at 8–9. The material was then dialyzed (Spectra/Por^®^1 dialysis membrane, 6–8 kDa, Fisher Scientific, Pittsburgh, PA, USA) against DI water for 5 days and lyophilized using a benchtop freeze dryer (Labconco FreeZone 4.5 L, Fisher Scientific, Pittsburgh, PA, USA). ^1^H NMR (Bruker Advance III HD 500 MHz, Bruker Scientific, Billerica, MA, USA) was used to confirm the methacrylate percentage as described previously [85].

### 2.2. Bone Particle Processing

Cancellous allograft bone (crushed cancellous bone from a 53-year-old male) was kindly provided by the Musculoskeletal Tissue Foundation (MTF) Biologics (Edison, NJ, USA). Crushed bone pieces were pulverized manually for 13 h by using a mortar and pestle set (JMD050, Deep Form, 50 mL, United Scientific Supplies Inc., Libertyville, IL, USA). A Mastersizer 3000 particle size analyzer (Malvern Panalytical Inc., Westborough, MA, USA) was used to study the particle size distribution.

### 2.3. Composite Bioink Preparation

Composite bioink formulation was prepared by dissolving MeALG powder, bone particles and photo initiator (LAP, lithium phenyl-2,4,6-trimethylbenzoylphosphinate, VWR International, Wayne, PA, USA) in phosphate-buffered saline (PBS, Fisher Scientific, Pittsburgh, PA, USA). First, LAP stock solution was prepared by dissolving 0.1% (*w*/*v*) LAP in phosphate buffered saline (PBS). Then, 3% (*w*/*v*) MeALG and 1% (*w*/*v*) bone particles were added into the LAP stock solution and kept under magnetic stirring for 5 days. To prepare 2 mL of ink, 0.06 g of MeALG and 0.02 g of bone particles were added into 0.1% LAP stock solution.

For in vitro culture studies, cell-laden composite bioinks were prepared by adding human mesenchymal stem cells (hMSCs, Lonza Walkersville Inc., MD, USA) in the composite ink formulation (~3 million cells per mL of ink formulation). For this purpose, hMSCs (passage 4, Lonza) were cultured in growth media (α-MEM (Gibco, Thermo Fisher Scientific LLC, Asheville, NC, USA) supplemented with 10% fetal bovine serum (FBS, Gibco, Thermo Fisher Scientific LLC, Asheville, NC, USA) and 1% penicillin-streptomycin (Gibco, Thermo Fisher Scientific LLC, Asheville, NC, USA) for ~80% confluency. Prior to dissolving polymer and bone particles, they were sterilized under ultraviolet germicidal irradiation for 2 h. LAP stock solution was filtered using a sterile syringe filter (0.22 µm, Sigma-Aldrich Inc., St. Louis, MO, USA). Subsequently, cells were mixed with ink under magnetic stirring for 20 min prior to 3D bioprinting.

### 2.4. Rheology

Kinexus Ultra + rheometer (Netzsch Instruments North America LLC, Burlington, MA, USA) was used to study the rheology of the bioinks. A parallel plate geometry (20 mm plate size and 0.7 mm gap size) was used. The viscosity of the ink was measured with respect to shear rate (0.01 to 1000 s^−1^). Strain sweep (0.5–300% at 1 Hz) and frequency sweep (0.1–100 Hz at 0.05% strain) tests were conducted to study the evolution of the elastic modulus (G′) and viscous modulus (G″). To investigate the light-induced crosslinking process, an optical kit (Netzsch) connected to a UV light source (Omnicure S2000 Excelitas Technologies, Chicago, IL, USA, 356 nm, 10 mW/cm^2^) was used. G′ and G″ were monitored with time (at 1 Hz). Light intensity was adjusted to represent the intensity during printing process (405 nm, 40 mW/cm^2^) according to the molar absorptivity spectrum of the photoinitiator (LAP) [45]. After 2 min of equilibrium, ink solution was exposed to the light (10 mW/cm^2^) for 20 min to fully crosslink the sample.

### 2.5. Optimization of 3D Bioprinting Parameters

In this study, we used a BIO X bioprinter (CELLINK LLC, Boston, MA, USA) with syringe-based print head and a 25G needle size with 0.25 mm internal diameter (Blunt End Dispensing Tip, 25G, Fisnar Inc., Germantown, WI, USA). A standard line test [45] was performed to evaluate the printability of the bioink formulations with respect to print pressure (100, 150, and 200 kPa) and speed (5–40 mm/s). Immediately after each strut (or line) was printed, it was exposed to UV light (405 nm, 2 mW/cm^2^) for 15 s to form a crosslinked hydrogel. Optical microscopy was used to capture the images of the struts, and ImageJ (NIH) was used to measure the strut width. A grid pattern (1 mm × 1 mm) was printed to evaluate the spatial uniformity of the printed struts. Here, the uniformity of the pores was investigated by drawing diagonal lines.

### 2.6. Characterization of Mechanical Behavior

The mechanical behavior of the composite hydrogels was evaluated using compression tests. Samples were fabricated in the form of disks (14 mm in diameter and 2 mm in height). Composite hydrogel samples were then weighted and soaked in PBS overnight to equilibrate their swelling. Samples were weighed again, and the swelling percentage of each sample was calculated using:(1)$$Swelling\;\%=\frac{w_{f}-w_{i}}{w_{i}}\times 100,$$
where *w_f_* and *w_i_* represent the weight at equilibrium swelling and post-printing, respectively. The equilibrated samples were then used for compression test. For this purpose, a Kinexus Ultra+ rheometer was used to apply an increasing normal force from 0.05 N to 15 N [45,95]. The gap was recorded to calculate the strain.

### 2.7. 3D Bioprinting of Composite Scaffolds

Cell-laden composite bioinks (3% MeALG with 1% bone particles) were printed on methacrylated glass slides [96] at optimized bioprinting parameters, such as 150–200 kPa at 20–30 mm/s for neat inks, and 150–230 kPa at 20–30 mm/s for composite inks. 3D scaffold designs were created by Autodesk^®^ Fusion 360™ and the 3D models were sliced with Slic3r in Repetier-Host to generate G-code files. A 2 mm × 2 mm grid scaffold composed of 4-layers (with a layer height of 150-µm and 500-µm offset between struts) was used for culture studies. Each bioprinted layer was partially crosslinked for 15 s to allow formation of a self-supporting layer (405 nm, 2 mW/cm^2^). Bioprinted scaffolds were exposed to light for 1 min to finalize the bioprinting process. Bioprinted cell-laden scaffolds were immediately transferred into non-treated 6-well plates, and 5 mL of growth media was added into each well.

### 2.8. In Vitro Studies

Cell viability studies were performed on 3D bioprinted scaffolds (4-layer grid scaffolds) using a Live/Dead staining kit (Invitrogen, Thermo Fisher Scientific LLC, Asheville, NC, USA) at culture days 1, 4, 7, 14, 21, and 28. In this assay, live cells were stained with calcein-AM dye (green, 0.5 μL/mL), and ethidium homodimer (red, 2 μL/mL) was used for staining dead cells. A confocal laser scanning microscope (TCS SP8 MP, Leica Microsystems Inc., Buffalo Grove, IL, USA) was utilized to capture cell images. Three images per scaffold were taken and transferred to ImageJ (NIH, Public Domain, Bethesda, MD, USA) software to analyze the cell viability by counting the number of live and dead cells.

To evaluate the osteogenic differentiation of the hMSCs, alkaline phosphatase (ALP) activity and alizarin red (AR) assay, as well as osteocalcin (OC) immunostaining were performed. For this purpose, 3D bioprinted scaffolds were cultured in growth media for one day, followed by culturing in osteogenic differentiation media for up to 28 days. Osteogenic differentiation media was prepared using high-glucose DMEM (Gibco, Thermo Fisher Scientific LLC, Asheville, NC, USA) supplemented with 100 nM of dexamethasone (Sigma-Aldrich Inc., St. Louis, MO, USA), 37.5 μg/mL of L-ascorbic acid (Sigma-Aldrich Inc., St. Louis, MO, USA), 10 mM of β-glycerophosphate disodium salt hydrate (Sigma-Aldrich Inc., St. Louis, MO, USA), 10% fetal bovine serum (FBS) (Gibco), and 1% penicillin-streptomycin (Gibco, Thermo Fisher Scientific LLC, Asheville, NC, USA). ALP activity was evaluated using QuantiChrom™ Alkaline Phosphatase Assay Kit (ALP assay Kit, Bio-Assay Systems, Hayward, CA, USA). For this purpose, 3 scaffolds per condition were collected at the desired culture time. Collected scaffolds were lysed with 0.25% Triton X-100 in DI water overnight. Then, lysate samples were reacted by adding a working solution prepared according to the protocol provided by the supplier. A plate reader (Infinite M200 Pro, Tecan Inc., Morrisville, NC) was used to read the absorbance at 405 nm. For AR staining assay, the collected scaffolds were fixed in 70% ethanol for 2 h. After DI water wash (3×), cells were stained with the AR staining kit (Sigma, St. Louis, MO, USA) at 4 °C overnight. Scaffolds were then washed with DI water several times to remove extra AR stain from scaffolds. After pictures of the scaffolds were taken, scaffolds were incubated in 10% cetylpyridinium chloride (Sigma, St. Louis, MO, USA) in sodium phosphate buffer (10 mM, pH 7, Sigma) overnight to extract the AR stain from cells. The collected solutions were scanned by a plate reader to read the absorbance at 562 nm to quantify calcium deposition. For OC immunostaining, scaffolds were collected at 14 days of culture. Cells were fixed by 4% formaldehyde solution (Sigma-Aldrich Inc., St. Louis, MO, USA) for 25 min, permeabilized with 0.25% Triton X-100 (Sigma-Aldrich Inc., St. Louis, MO, USA) for 1 h, and subsequently incubated in a blocking solution (10% goat serum (Thermo Fisher Scientific LLC, Asheville, NC, USA) in PBS) for 3 h at room temperature. The OC primary antibody (1:200, monoclonal mouse, Invitrogen, Thermo Fisher Scientific LLC, Asheville, NC, USA) in staining solution (3% bovine serum albumin + 0.1% Tween-20 + 0.25% Triton X-100) was prepared and used as the primary staining of the cells. Cells were incubated in primary staining solution for 48 h at 4 °C. Later, cells were stained by Alexa Fluor 488 rabbit anti-mouse secondary antibody (1:100, Invitrogen, Thermo Fisher Scientific LLC, Asheville, NC, USA) in a staining solution for 24 h. In addition, phalloidin (rhodamine phalloidin, Invitrogen, Thermo Fisher Scientific LLC, Asheville, NC, USA) and DAPI (Thermo Fisher Scientific LLC, Asheville, NC, USA) were used to stain the cells for imaging F-actin and cell nuclei, respectively. Confocal and multiphoton microscopy (TCS SP8 MP, Leica Microsystems Inc., Buffalo Grove, IL, USA) is used to image the cells.

### 2.9. Statistics

The data were analyzed by Minitab software (Version:20.3.0, Minitab, LLC, State College, PA, USA) and presented as mean ± standard deviation for n ≥ 3 samples. The analysis of variance (ANOVA) with Tukey and a 95% level of confidence was used for comparison between conditions. A *p*-value < 0.05 was considered statistically significant.

## 3. Results and Discussion

### 3.1. Composite Bioink Formulation and Characterization

In this study, human allograft bone particle containing composite bioinks are developed for material extrusion-based 3D bioprinting, also known as direct ink writing (DIW), to bioprint cell-laden composite hydrogel scaffolds for bone tissue development (Figure 1). Methacrylated alginate (MeALG), with ~80% methacrylate (Me) functionalization, was synthesized and used as the photocurable hydrogel component of the composite bioink. Human allograft bone particles are mixed with MeALG and dissolved in PBS to form a composite ink formulation (Figure 1). Based on our previous studies [45] and initial screening tests, the composition of the MeALG is set to 3% (*w*/*v*). Considering the cell-laden nature of the bioinks, bone particle concentration was limited to 1% (*w*/*v*). As received human bone allograft tissue (in the form of large chips) is pulverized up to 13 h to form uniform particles with ~16 µm average particle size. Figure 2 shows the SEM image of the particles (Figure 2A) and the evolution of particle size distribution with grinding time (Figure 2B). After 1 h of grinding, the average bone particle size is 189 µm, with a broader particle size distribution composed of a main peak (50–580 µm range) with a wide tail towards lower particle sizes (0.5–50 µm). The main particle distribution peak gradually shifts to lower particle sizes with increasing grinding time, and a sharper peak emerges at the range of 0.4 to 46 µm (base width) corresponding to a mode equal to 16.4 µm after 13 h grinding.

The change in ink viscosity with shear rate for neat and composite inks are given in Figure 3A. The shear viscosity of the MeALG bioink significantly increased with the addition of bone particles, such that viscosity values at low shear rates (0.01 s^−1^), approaching to zero shear viscosity, increased from ~7 to ~30 Pa·s. Both ink formulations show shear thinning behavior indicated by the significant decrease in shear viscosity with increasing shear rate. Shear thinning behavior of the MeALG is known to be associated with chain entanglement [97], which resists ink flow at low shear rates. The degree of shear thinning is higher for composite bioinks as expected. This is due to the presence of particle-particle interactions, which enhances resistance to flow indicated by high zero shear viscosity values in filled polymer solutions [98]. Particle-particle interactions are destroyed at higher shear rates leading to a significant decrease in ink viscosity [99]. Figure 3B illustrates the change in elastic modulus (*G′*) and viscous modulus (*G″*) with increasing shear strain (0.5–300% at 1 Hz). Both ink formulations behaved like a liquid indicated by *G″* > *G′*, yet the difference between *G′* and *G″* significantly reduced for composite inks, supporting the observed increase in viscosity in Figure 3A. Frequency sweep tests (0.1–100 Hz at 0.05% strain) shown in Figure 3C indicate increasing *G′* and *G″* with increasing frequency. The frequency dependency of the *G′* confirms the viscous behavior of the inks.

MeALG solution forms crosslinked hydrogels via radical polymerization when exposed to light in the presence of a photoinitiator. The crosslinking density, and hence stiffness of the hydrogel can be controlled by the % methacrylation, initiator concentration and light exposure time [45,85]. To study the effect of particles on photocuring kinetics, we monitored the change in *G′*, *G″* and phase angle (*δ*) under light exposure (Figure 4). Figure 4A shows the results for neat MeALG ink. Initially, the ink behaves like a liquid with *G″* >> *G′*. When the ink is exposed to light, *G′* increases significantly and becomes larger than *G”* due to start of the crosslinking reaction with a gel point (at 126 s) defined at *G″* = *G′*. A significant drop in *δ* is also an indication of the gelation [100]. Both *G″* and *G′* reach to an equilibrium indicating the competition of the crosslinking reaction. The composite ink behaves similarly, and the gel point is similar (127 s). In summary, the presence of 1% bone particles does not affect the crosslinking behavior of the composite inks.

### 3.2. Mechanical Properties

The mechanical behavior, stiffness, or Young’s modulus (*E*) of the composite hydrogels are studies using compression tests (Figure 5). Although *G′* indicates the elastic modulus of the hydrogels (Figure 4), hydrogels should be equilibrated in PBS to determine the actual stiffness under in vitro culture conditions (Figure 5A). Our results show that the % swelling decreases from 84% to 72% for composite hydrogels. In good agreement with the swelling results, *E* increases from 21 ± 6.2 kPa (for neat gel) to 51 ± 7.1 kPa (for composite gel). Note that covalently crosslinked MeALG is known to be stable under in vitro conditions; however, cell-mediated degradation can be achieved by using enzymatically degradable crosslinkers [85].

### 3.3. Printability

Standard line test study is performed to investigate the printability of the neat and composite MeALG. In general, print strut size (width) increased with increasing print pressure at a constant print speed, whereas strut size decreased with increasing print speed at a constant print pressure (Figure 6). We were able to print uniform struts with as low as 600 µm (at 100 kPa and 20 mm/s) and 700 µm (at 100 kPa and 30 mm/s) for composite and neat ink formulations.

Grid patterns are also printed to confirm the printability of both inks. Here, the print speed needs to be adjusted slightly to create the grid patterns with uniform struts and gap (or pore) (Figure 7). For both ink formulations, we observe collapse of pores at higher print pressures and low print speeds due to deposition of excess ink. For lower pressures and higher speeds, struts are more pronounced with clear definition of pores, such that circular pores become squares (Figure 7C,D). Pluronic is also used as a control as it is known to be easily printed to form self-supporting structures [48]. The diagonal line of the square shape gap is measured to determine print quality [101]. Our results show that the length of the diagonal line decreases with increasing print pressure and the square shape converges to a rounded shape (Figure 8). Here, we would like to note that square shape is preserved at higher speeds, yet the printed struts become thinner, making them faster to dry out prior to completion of the print job, potentially leading to cell death as discussed below. Therefore, it is necessary to compromise the perfect square pores for rounded pores to achieve bioprinting reproducible scaffolds with controlled shape and high cell viability.

### 3.4. In Vitro Studies

In this section, hMSC viability (Figure 9) and differentiation (Figure 10) results are presented for culture times up to 28 days. It is well known that MeALG is blank to cells as it does not contain inherent bioactivity toward cells and needs to be functionalized with bioactive cues to promote specific biological responses. MeALG is usually modified with integrin binding arginine-glycine-aspartic acid (RGD)-peptides to promote cell survival for matrix tethering cells, such as the adult stem cells used in this study [85,94,96]. Methacrylate (Me) pendant groups allow chemical tethering of bioactive molecules containing cysteine groups mainly through a Michael-type addition reaction. Here, we functionalized MeALG with RGD-peptide following the protocol developed previously to enhance stem cell-matrix adhesion [85,94,96].

Our results show that % cell viability is ≥89% for all time points, except for neat scaffolds at day 1 (85 ± 2%), yet the data is not significantly different than that of composite day 1 (89 ± 0.1%) (Figure 9B). We observed a slight increase at day 21 (97 ± 0.1% (*N*) and 96 ± 0.1% (*C*)) and day 28 (91 ± 0.1% (*N*) and 93 ± 0.1% (*C*)), which could be due to delayed proliferation within scaffolds. Bioprinted scaffolds show a slight decrease in thickness with culture time which is attributed to detachment of the scaffolds from the glass slide as observed visually during culture. We believe that this led to the collapse of some layers or breakage of the sample for prolonged culture times.

Osteogenic differentiation studies are performed up to 28 days of culture, using neat and composite scaffolds, in osteogenic media, and growth media condition is used as a control. Our results show that ALP activity (normalized activity with respect to activity recorded at culture day 1) increases significantly with culture time for both neat and composite scaffolds (Figure 10A). For each culture day, ALP activity is significantly higher for the composite scaffolds, indicating significantly enhanced osteogenic differentiation in the presence of bone particles. Note that hMSCs express ALP without differentiation, and it should not be used as a sole indicator of osteogenic differentiation. Normalized ALP activity is very low in growth media and does not change with culture time for neat scaffolds, whereas the activity increased with culture time for composite scaffolds such that 3.5× increase is recorded at day 28 (as compared to day 1). This is much smaller than the activity observed at day 28 (48×) and even at day 7 (9×) in differentiation media. The majority of the hMSCs (~95%) stained positive for osteocalcin (OC) within composite scaffolds as compared to neat scaffolds (~70%) (Figure 10B). No staining is observed for samples cultured in growth media (results not shown). Alizarin red (AR) staining is used to evaluate calcium deposition, which is an indicator for osteogenic differentiation of hMSCs. AR staining for composite scaffolds is significantly darker as compared to neat scaffolds (Figure 10C–D). Some (significantly dim in color) AR staining is observed for composite scaffolds cultured in growth media indicating some calcium deposition. In addition, AR assay is performed to quantitatively measure AR activity (Figure 10E). Confirming our qualitative assessments, AR expression is significantly higher for composite scaffolds when compared with neat scaffolds for both growth and OC media, with significantly higher AR expression in differentiation media. These results are in good agreement with ALP activity and OC staining results. Overall, our results clearly indicate that the presence of bone particles significantly enhanced osteogenic differentiation of hMSCs within bioprinted composite scaffolds.

Decellularized human bone particles sustain the bioactivity of the native human bone tissue including biominerals as compared to commonly used digested bone tissue, which requires decellularization and demineralization [58]. Thus, combining human bone particles with a photocurable hydrogel is a novel approach to formulate bone mimetic bioinks for extrusion-based bioprinting to fabricate scaffolds for bone tissue engineering. In this study, we report a visible light curable composite bioink formulation composed of 3% (*w*/*v*) MeALG and 1% (*w*/*v*) bone particles. Although bioceramics are commonly used to fabricate hydrogel-based composite bone inks, these only target the biomineral component and lack the ECM of the bone tissue. Composite bone inks focusing on human bone allograft particles provide a more complete bone mimetic microenvironment, yet they are rarely reported in the literature. Compared to a similar approach reported previously, which utilized significantly higher concentrations of MeALG (5–15%, *w*/*v*) and bone particles (10–75% *w*/*v*) [69], we are able to formulate bone mimetic inks with significantly lower amounts of ingredients providing a much more feasible path for clinical applications—considering the cost and availability of the ingredients. Our current study clearly shows that bone particles can be used as a reinforcement to adjust the rheology of the bioink and, hence, printability, as well as to enhance the mechanical properties of the bioprinted hydrogels. Note that covalently crosslinked MeALG hydrogels are stable within the time frame of the in vitro experiments and should not be degrading in the presence of cells, yet degradation can be included by using an enzymatically degradable peptide crosslinker [85]. Our results show that bone particles do not interfere with the photocrosslinking step making it easy to bioprinting and potentially use it as an injectable formulation. Most importantly, the presence of bone particles significantly enhances hMSC differentiation towards osteogenic phenotype, such that hMSCs within composite hydrogels show significantly higher ALP activity, AR staining and activity, and osteocalcin staining as compared to the cells within neat hydrogels.

## 4. Conclusions

In this study, we report a novel photocurable composite bioink composed of methacrylated alginate with human bone allograft particles for 3D bioprinting of bone scaffolds. Composite inks show higher low shear viscosity but enhanced shear thinning due to presence of bone particles. Incorporation of bone particles leads to a decrease in swelling and an increase in stiffness of the composite hydrogels as compared to neat hydrogels. Standard line tests and grid patterns are used to optimize the print pressure and speed to fabricate uniform scaffolds. In vitro culture studies up to 28 days reveal high cell viability (~90%) for hMSCs when bioprinted within neat or composite bioinks. Differentiation studies confirm significantly high alkaline phosphatase activity, calcium deposition, and osteocalcin expression for hMSCs within bioprinted composite hydrogels as compared to neat hydrogels. Overall, our results confirm that our composite bioinks have a significant potential to create scaffolds for bone tissue regeneration.

## Figures and Tables

**Figure 1 polymers-14-03788-f001:**
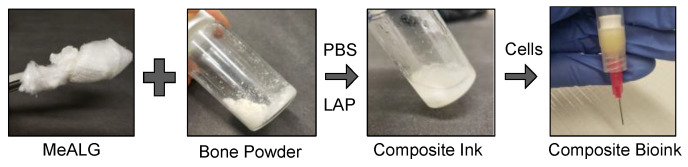
Preparation of composite bioinks. Pictures showing MeALG, bone powder, and composite ink solution in PBS containing photoinitiator (LAP). After addition of hMSCs, composite bioink is transferred into a syringe for bioprinting.

**Figure 2 polymers-14-03788-f002:**
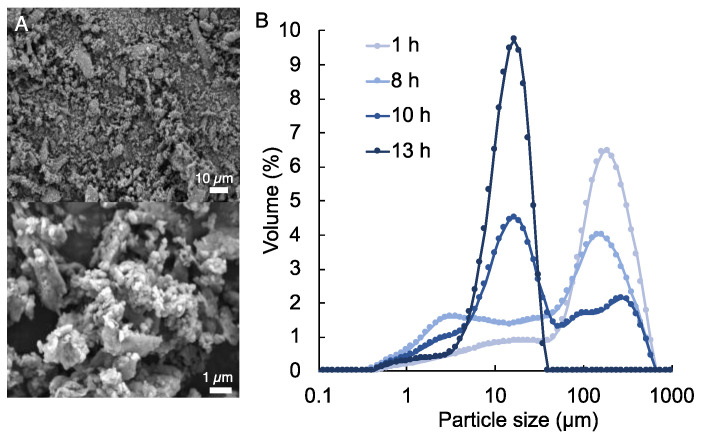
(**A**) SEM images of the bone particles that are pulverized for 13 h. (**B**) Plot showing the particle size distribution for pulverized (1–13 h) bone particles.

**Figure 3 polymers-14-03788-f003:**
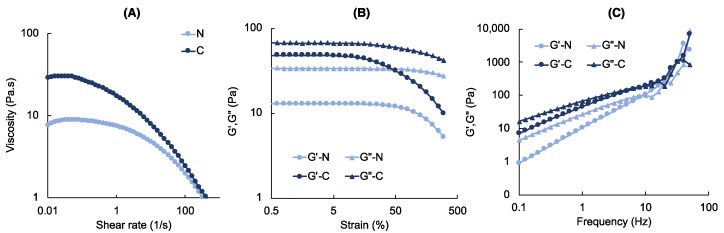
Rheology of the neat (*N*) and composite (*C*) inks. (**A**) Change in ink viscosity with shear rate. (**B**–**C**) Plots showing elastic (*G′*) and viscous (*G″*) modulus under (**B**) strain and (**C**) frequency sweep.

**Figure 4 polymers-14-03788-f004:**
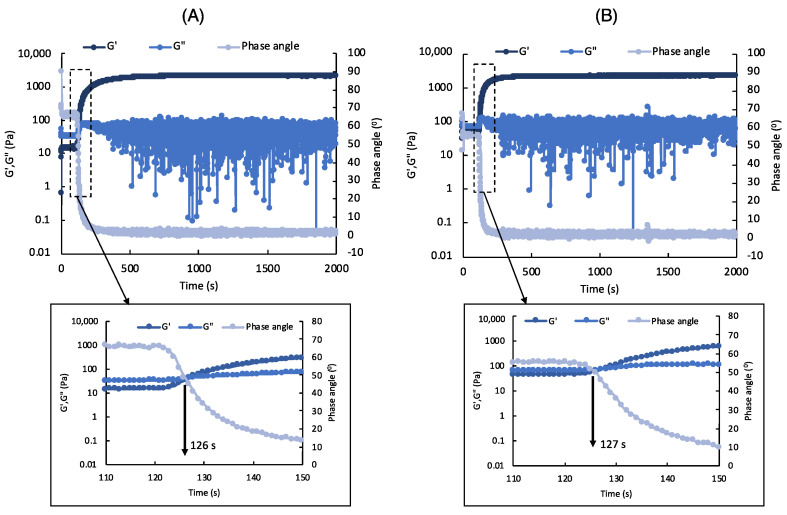
Rheological studies to investigate crosslinking behavior of the neat (*N*) and composite (*C*) inks. (**A**–**B**) Change in elastic modulus (*G′*), viscous modulus (*G”*), and phase angle with time for *N* (**A**) and *C* (**B**) inks. Light is turned on at 120 s for 20 min. The plots at the bottom show the zoomed in regions indicated by the dotted lines in the top plots.

**Figure 5 polymers-14-03788-f005:**
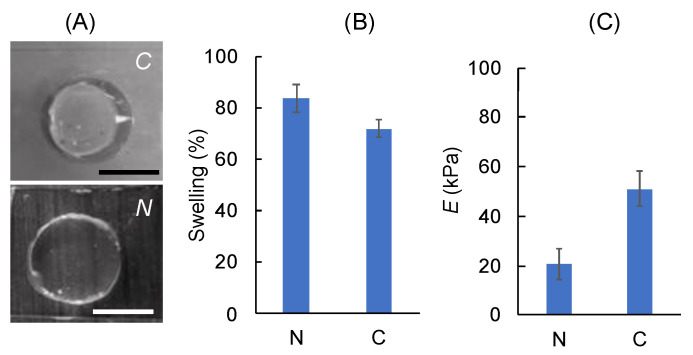
Mechanical characterization of the composite (*C*) and neat (*N*) hydrogels. (**A**) Pictures showing the hydrogels after swelling. Scale bars are 1 mm. (**B**) Percent swelling and (**C**) elastic modulus (*E*) of the hydrogels. Data are presented as mean ± std. deviation for n = 3 samples.

**Figure 6 polymers-14-03788-f006:**
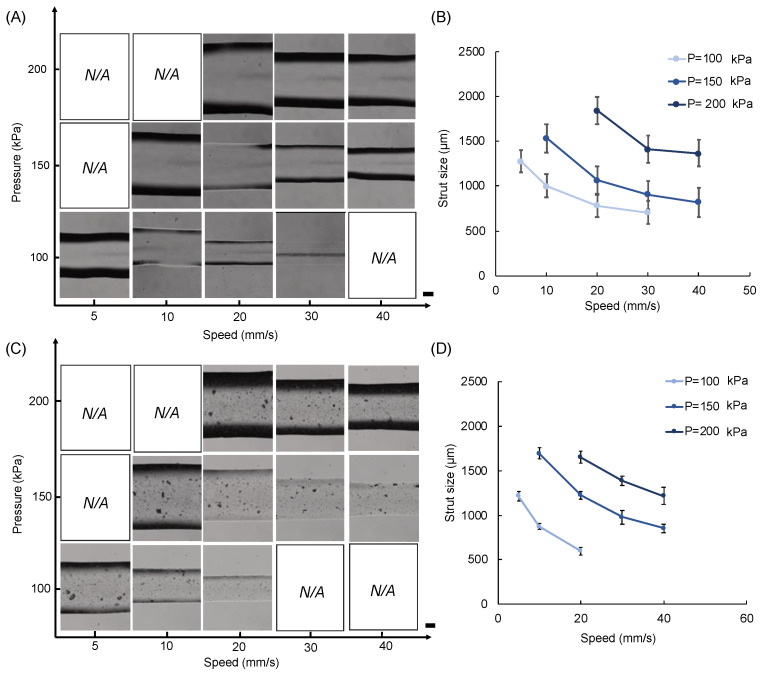
Standard line test results for (**A**,**B**) neat (*N*) and (**C**,**D**) composite (*C*) inks. Microscope images showing the 3D printed lines (struts) (**A**,**C**) and plots showing the corresponding measured strut width values (**B**,**D**). Scale bars are 200 microns. Data are presented as mean ± std. deviation for n = 3 samples with 3 measurements for each sample.

**Figure 7 polymers-14-03788-f007:**
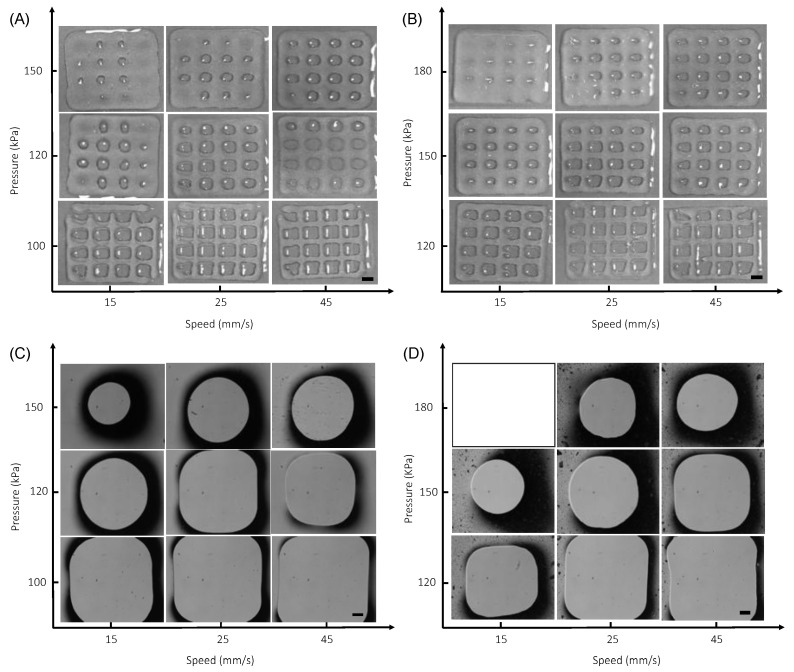
Printability studies via printing grid patterns using (**A**,**C**) neat (*N*) and (**B**,**D**) composite (*C*) inks. (**A**,**B**) Pictures showing 3D printed grid patterns with respect to print pressure and speed. Scale bars are 1 mm. (**C**,**D**) Bright field images showing a representative pore for each scaffold. Scale bars are 200 microns.

**Figure 8 polymers-14-03788-f008:**
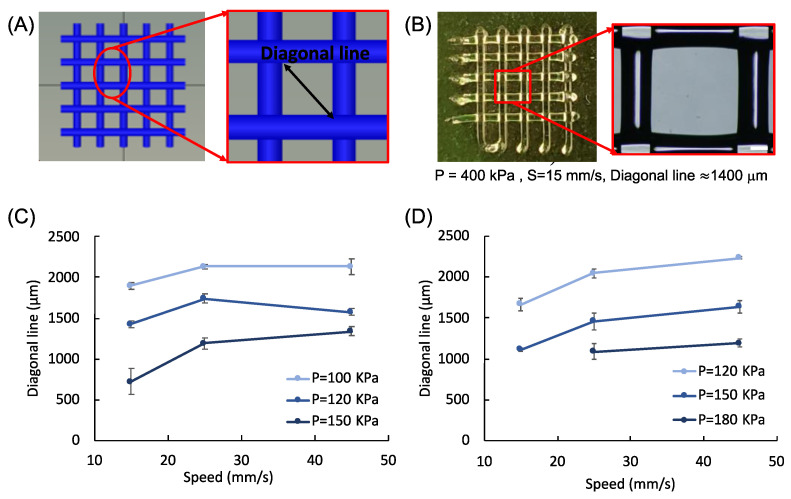
Characterization of printability. (**A**) Images showing the grid design and measurement of the diagonal line. (**B**) Results from pluronic inks. Scale bar is 200 microns. (**C**,**D**) Diagonal line length with respect to print pressure and speed for neat (*N*) and composite (*C*) scaffolds.

**Figure 9 polymers-14-03788-f009:**
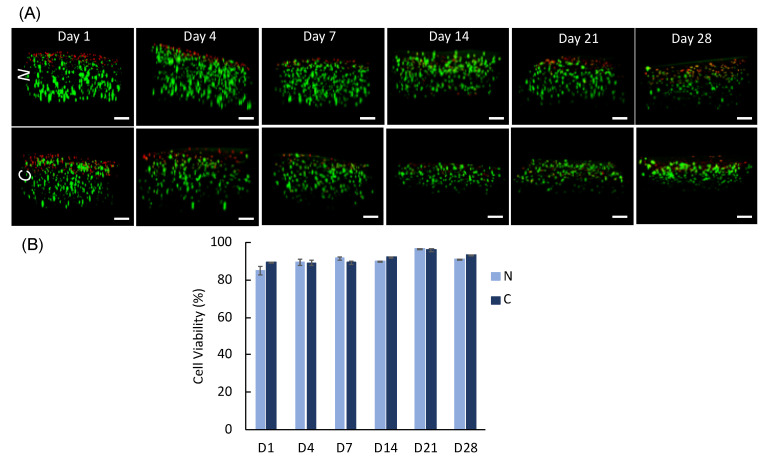
Cell viability studies. (**A**) Confocal images of hMSCs showing cross-sectional views of neat (top) and composite (bottom) scaffolds. Green and red cells indicate live and dead cells, respectively. Scale bars are 200 µm. (**B**) Plot showing percent cell viability with culture day for hMSCs within bioprinted neat (*N*) and composite (*C*) scaffolds. Data are presented as mean ± std. deviation for n = 3 samples with 3 measurements for each sample.

**Figure 10 polymers-14-03788-f010:**
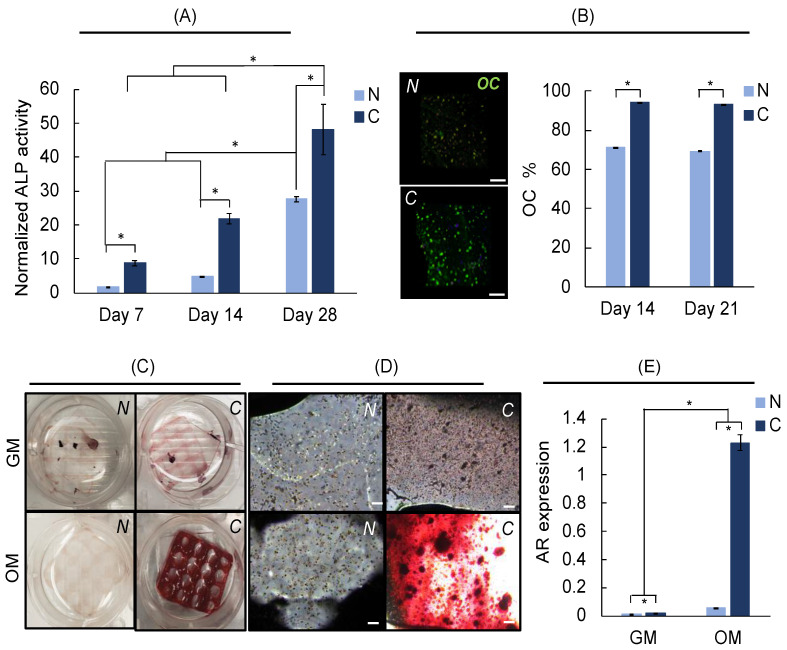
Differentiation results of hMSCs within bioprinted neat (*N*) or composite (*C*) hydrogels cultured up to 28 days in growth media (GM) or osteogenic differentiation media (OM). (**A**) Normalized ALP activity (with respect to day 1). (**B**) Confocal images of hMSCs showing osteocalcin (OC, green) staining at day 14 (scale bars are 200 microns) and plot showing percent OC staining for hMSCs. (**C**) Pictures of scaffolds at day 21, red color indicating alizarin red (AR) staining. (**D**) Microscope images of the scaffolds corresponding to the scaffolds in (**C**). Scale bars are 200 microns. (**E**) Plot showing AR expression at day 21. * indicates *p* < 0.05 for n = 3 samples per group.

## Data Availability

The data presented in this study are available on request from the corresponding author.

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
