# Peer review of "Cell-Laden Composite Hydrogel Bioinks with Human Bone Allograft Particles to Enhance Stem Cell Osteogenesis"

_polymers, 2022, doi:10.3390/polym14183788_

Round 1

Reviewer 1 Report

Bone defects have been a great problem facing public health. The manuscript ‘polymers-1892653’ has reported a composite bioink formulated by incorporating human allograft bone particles in methacrylated alginate formulation to enhance adult human mesenchymal stem cells (hMSC). This article is potentially interesting to readers from both from the academic field and clinical applications. However, the following revisions need to be made before the final publication.

 (1)   The paper should be reorganized with a more logical order, instead of describing a series of figures;

(2)   The advantage of the present method compared with other commonly used methods should be provided;

(3)   The detailed structure of the hydrogel loaded with cells should be provided, either by TEM or SEM;

(4)   The mechanical properties of the hydrogel load with cells after differentiation should be provided;

(5)   The spelling of words and the grammar needs to be improved.

Reviewer 2 Report

The authors have submitted an interesting article entitled "Cell-Laden Composite Hydrogel Bioinks with Human Bone Allograft Particles To Enhance Stem Cell Osteogenesis" which deals with imparting human allograft bone particles in methacrylated alginate formulation to enhance adult human mesenchymal stem cell (hMSC) osteogenesis. The manuscript reads well overall, although it will need a spelling and style check. I suggest this article be published after a minor revision.

Comments:

 1- First of all, I would like to recommend authors to design a better “Graphical Abstract” (GA) for this study to better show the whole story in a simple and informative manner. The proposed GA is better to comprise all material modification chemistry (a simple schematic), along with some SEM images, in vitro study, and so forth.

 2- Please explain more why you have chosen to use “methacrylic anhydride” over “glycidyl methacrylate” to conjugate the methacryloyl group to the alginate backbone. The following paper may be helpful: https://doi.org/10.3389/fmats.2019.00263

3- The novelty statement of the article poorly represents the work and needed to be developed to highlight the importance of this work and how it is different from previously published reports.

4-Some of the references in the introduction part are too old (e.g., 2001, 2004). A myriad of research bodies has been published in recent years and you can find similar concepts and cite them in your paper rather than around 2 decades old references. You need to update references from recently published articles (last 3 or 5 years). Please read, add valuable information, and cite the following key papers: “Chitosan-based inks for 3D printing and bioprinting, DOI            https://doi.org/10.1039/D1GC01799C” to introduction line 76 and “Alginate-Based Bioinks for 3D Bioprinting and Fabrication of Anatomically Accurate Bone Grafts”.

5- Please provide mass loss (degradation profile) of the material in physiological condition (PBS or Media @ 37°C)

Round 2

Reviewer 1 Report

The authors have adequately addressed all my concerns, and the paper should now be accepted after a little more attention is given to grammatical errors.